# IL13 May Play an Important Role in Developing Eosinophilic Chronic Rhinosinusitis and Eosinophilic Otitis Media with Severe Asthma

**DOI:** 10.3390/ijms222011209

**Published:** 2021-10-18

**Authors:** Hideyasu Shimizu, Masamichi Hayashi, Hisayuki Kato, Mitsuru Nakagawa, Kazuyoshi Imaizumi, Mitsushi Okazawa

**Affiliations:** 1Toshiwakai Clinic, Toshiwakai, Nagoya 460-0022, Japan; hyde1040@icloud.com; 2Department of Respiratory Medicine, Fujita Health University School of Medicine, Toyoake 470-1192, Japan; michi@fujita-hu.ac.jp (M.H.); jeanluc@fujita-hu.ac.jp (K.I.); 3Department of Otolaryngology-Head and Neck Surgery, Fujita Health University School of Medicine, Toyoake 470-1192, Japan; katoq@fujita-hu.ac.jp; 4Okazaki Medical Center, Department of Diagnostic Pathology, Fujita Health University School of Medicine, Okazaki 444-0827, Japan; nakaga1@fujita-hu.ac.jp; 5Department of Respiratory Medicine, Daiyukai General Hospital, Daiyukai Health System, Ichinomiya 491-8551, Japan

**Keywords:** benralizumab, dupilumab, mepolizumab, severe asthma, eosinophilic chronic rhinosinusitis, eosinophilic otitis media

## Abstract

A woman in her 50s was a super responder to benralizumab administered for the treatment of severe bronchial asthma (BA) with eosinophilic chronic rhinosinusitis with nasal polyp (ECRS) and eosinophilic otitis media (EOM). She exhibited the gradual exacerbation of ECRS/EOM despite good control of BA approximately 1 year after benralizumab initiation. Therefore, the treatment was switched to dupilumab, and the condition of the paranasal sinuses and middle ear greatly improved with the best control of her asthma. The patient reported that her physical condition was the best of her life. However, she developed a pulmonary opacity on chest computed tomography after 6 months. Histological examination of the lung parenchyma and cell differentiation of the bronchoalveolar lavage fluid indicated atypical chronic eosinophilic pneumonia, and treatment was switched to mepolizumab. Similarly to the period of benralizumab treatment, exacerbation of ECRS/EOM reduced her quality of life approximately 10 months after the administration of mepolizumab. Dupilumab was again introduced as a replacement for mepolizumab. The clinical course and consideration of the interaction between inflammatory cells led us to speculate that interleukin-13 could play a key role in the development of ECRS/EOM with severe BA.

## 1. Introduction

Bronchial asthma is a chronic airway disorder in which the interaction of a variety of cells induces chronic inflammation and remodelling of the airway [1]. Severe asthma is defined as uncontrolled condition despite adequate treatment with high dose of corticosteroids; it exerts not only physical but also socioeconomic burden [1]. Eosinophilic chronic rhinosinusitis (ECRS) and eosinophilic otitis media (EOM) drew attention in the 1990s as new disease entities that were difficult to treat. Patients with ECRS exhibit multiple nasal polyps on both sides with severe eosinophils, olfactory disturbance, and paranasal sinusitis with greater fluid accumulation in the ethmoid sinuses than other sinuses [2]. EOM occurs predominantly in women in their 50s and complicates with ECRS and severe asthma [3]. The accumulation of highly viscous effusion and granulation with eosinophil infiltration in the middle ear causes bulging of the eardrum and frequently results in perforation, causing a hearing disturbance [4]. Therefore, a combination of these conditions with severe bronchial asthma deteriorates quality of life (QOL).

Eosinophils play a key role in the development of ECRS, EOM, and severe bronchial asthma [1,2,3]. Since interleukin 5 (IL5) is a major cytokine that activates eosinophils to induce eosinophilic inflammation in bronchial asthma, ECRS, and EOM, anti-IL5 mono-clonal antibodies were introduced as a treatment modality for these intractable disorders [5]. In a previous report, we showed the rapid and remarkable efficacy of benralizumab for the treatment of severe asthma with intractable ECRS/EOM [6]. The clinical course of this patient over the past 3 years, however, was not uneventful and biologic treatment had to be switched. In the current report, we describe the difficulty in treating severe asthma with ECRS/EOM in this particular patient and discuss the cellular interactions underlying type 2 inflammation.

## 2. Case Presentation

A woman in her 50s, who was a super responder to benralizumab for the treatment of severe BA with ECRS/EOM, was complaining of a slight and unusual feeling in her throat and chest on approximately 10 months after treatment with benralizumab.

Since respiratory resistance (Rrs;R5 and R20), as measured by impulse oscillometry [7], increased (Table 1; May 2019), long-acting muscarinic antagonist (LAMA; Spiriva Respimat^®^ 2 puffs daily) was added to her inhaled corticosteroid/long-acting β agonist (ICS/LABA;Simbicort200^®^ 8 times daily) and leukotriene receptor antagonist (LTRA; montelukast 10 mg daily). Her respiratory symptoms and increased Rrs recovered completely (Table 1; October 2019). Inhalation therapy was, then, switched to a triple therapy (ICS/LABA/LAMA; Trelegy^®^) once daily. At approximately 13 months of treatment with benralizumab, nasal obstruction and post nasal dripping (PND) gradually increased and a bulging of the eardrum was observed, despite alternative-day administration of celestamine^®^ (compounding agent of 0.25 mg of betamethasone and 2 mg of d-chlorpheniramine maleate). Myringotomy was necessary, and sticky yellowish mucous was aspirated, although the stickiness was much less than that during biologics-free period [6]. Since nasal obstruction with olfactory disturbances, nasal voice, PND and otorrhea with hearing loss were worsening, benralizumab was discontinued and subcutaneous injection of 300 mg of dupilumab was added to Trelegy^®^ and montelukast (December 2019). Subsequently, nasal obstruction, PND, nasal voice and otorrhea rapidly and completely diminished and the oral administration of celestamine^®^ was no longer necessary (Table 1; January 2020). Her olfactory disturbances and hearing loss completely recovered. Her asthma control test (ACT) result, mini-asthma quality of life questionnaire (mAQLQ) score, ratio of forced expiratory volume in 1 s vs. forced vital capacity (FEV_1_/FVC%), respiratory resistance (Rrs) and resonance frequency (Fres) were unchanged (Table 1; January 2020).

The patient reported her asthma control, nasal and ear conditions were the best of her life. Although ACT, mAQLQ and pulmonary function tests were unchanged for 5 months following the administration of dupilumab, an ICS (100 μg of ARNUITY^®^ once daily) was temporally added to Trelegy^®^ since the patient reported a slightly unusual feeling in the chest. As this symptom continued for next 1 month, despite the increased ICS, chest computed tomography (CT) was performed.

Ground glass opacities and patchy consolidations with air-brochogram were observed in the sub-pleural area of upper lobes (Figure 1A). The eosinophil count in the peripheral blood was dramatically increased (Table 1; June 2020).

The results of bacteriological tests, including thoe for fungal and acid-fast bacilli, were negative and anti-neutrophil cytoplasmic antibody (ANCA) and sialylated carbohydrate antigen (KL-6) were not elevated; thus, bronchoalveolar lavage and transbronchial lung biopsy were performed. The cell differentiation of the bronchoalveolar fluid (BALF) was as follows: eosinophil 7%, neutrophil 0%, lymphocyte 31%, histiocyte 58%, mast cell 4%. The CD4/CD8 ratio of her T cell subset was 0.6. The result of the drug-induced lymphocyte stimulation test for dupilumab was negative. Hematoxylin-eosin staining of biopsy specimens showed hyalinous thickening of the alveolar wall with intra-alveolar buds of granulation tissue (Figure 1B), infiltration of lymphocytes, plasma cells, eosinophils and histiocytes in the lung parenchyma (Figure 1C). Dupilumab was discontinued and prednisone of (25 mg/day) was started with Trelegy^®^ and montelukast^®^. The pulmonary opacities were promptly disappeared in two weeks, and prednisone dose was tapered before discontinuation over the next 5 months. During this time, the good control of BA, ECRS and EOM continued. One month after prednisone administration (July 2020), mepolizumab was started. The eosionophil counts in the peripheral blood decreased (Table 1; October 2020), but not to the same extent as during the period of benralizumab treatment. After 6 months of treatment with mepolizumab, FEV_1_/FVC%, Rrs, ACT and mAQLQ did not change, with no nasal and ear symptoms as had been observed during the period of dupilumab treatment (January 2021). However, nasal obstruction, PND, nasal voice and otorrhea gradually worsened and olfactory disturbance and hearing loss developed 11 months after administration of mepolizumab, as was the case with benralizumab.

Endoscopic examination showed a small nasal polyp in the left olfactory cleft (Figure 2A) and edematous middle turbinate covered with whitish nasal discharge (Figure 2B). The ear canal and eardrum were covered with a copious amount of dried discharge (Figure 2C).

Since the patient’s QOL deteriorated because of her nose and ear symptoms, mepolizumab was switched again to dupilumab (August 2021). Subsequently, the symptoms of nose and ear were completely disappeared as in the previous dulilumab treatment. Endoscopic observation 1 month after dupilumab treatment revealed no olfactory cleft polyps, no inflammation in middle turbinate, and the normal appearance of the ear canal and eardrum (Figure 3D–F) was observed. Olfactory disturbance and hearing loss had completely recovered. The CT findings of paranasal sinuses were striking (Figure 3).

In a previous report [6], we showed that the application of benralizumab, along with endoscopic nasal polypectomy and sinus opening surgeries, greatly decreased the amount of fluid in the bilateral ethmoidal sinuses, although the amount of fluid in bilateral maxillary sinuses was comparable to that in the biologic-free condition (Figure 3A,B). The accumulation of fluid in ethmoidal and maxillary sinuses during benralizumab and mepolizumab was similar (Figure 3B,C). The fluid accumulation in both sinuses had almost completely disappeared 1 month after treatment with dupilumab (Figure 3D), and eosinophil count in peripheral blood remained normal (Table 1; September 2021). Currently, the patient is in the best control of her BA, with normal sense of smell and hearing, as in a previous period with dupilumab.

## 3. Discussion

The clinical course of the patient was complicated. Over the past 3 years, our patient was treated with benralizumab, dupilumab, mepolizumab and dupilumab again, in sequence. Since the initial treatment with benralizumab showed rapid and dramatic improvement in asthma control as well as nose and ear symptoms, we believed that the elimination of eosinophils by benralizumab could be the best treatment for these conditions. However, ECRS/EOM gradually exacerbated approximately 1 year after treatment with benralizumab, and the treatment was switched to dupilumab. Although the best control of asthma was achieved, and the nasal nose and ear symptoms were greatly improved after 5 months, pulmonary infiltration with eosinophilia appeared, and discontinuation of dupilumab with prednisone treatment was necessary. However, following treatment with mepolizumab for the next 11 months, the nose and ear symptoms gradually aggravated to impair QOL to a similar degree as in the period of benralizumab treatment. Therefore, a second round of dupilumab was started. To elucidate the clinical course of our case, it is necessary to briefly review the cellular interaction in the adaptive and innate immune systems. The outline of the cross-talk between immune cells involved in airway inflammation is depicted in Figure 4.

Inhaled allergens, pathogens or air pollutants are captured by pattern-recognition receptors such as toll-like receptors (TRLs) expressed on the variety of cells including epithelial cells, innate lymphoid cells (ILC), neutrophils and the antigen-presenting cells (APC) such as dendritic cells (DC), macrophages (Mϕ) and mast cells (MC) [8]. In the allergic inflammation, activated DC plays an essential role in adaptive immunity by capturing antigens, presenting antigens to naïve CD4 T cells (Th0) through class II major histocompatibility complexes (MHCII) and T cell receptors (TCR), and by differentiating Th0 into type 2 helper T cells (Th2) by multiple factors including IL4 [9,10,11]. Th2s release IL4 and IL13 to differentiate B cells into plasma cells to produce allergen-specific IgE antibodies and also release IL5, which is a major cytokine responsible for proliferation, maturation, migration, activation and survival of eosinophils [12]. In contrast, Group 2 ILCs (ILC2s) in the innate immune system are activated by IL25, 33 and thymic stromal lymphoprotein (TSLP), which are discharged from epithelium or macrophages [13] without antigen presentation since ILCs lack TCR. Activated ILC2s release a large amount of IL5 and IL13 and smaller amount of IL 4 [14,15].

As illustrated by Figure 4, type 2 inflammation (red arrows), which involves IL4, IL5, IL9 and IL13 is organized by Th2s in acquired immune system and ILC2s in innate immune system. The activation of Th2 and ILC2 and cross-talk between these cells orchestrates type 2 inflammation and contributes to the development of asthma pathophysiology, such as increased mucous production, over production of IgE, subepithelial fibrosis, airway wall remodeling and airway hyperresponsiveness [16]. Other types of inflammation, such as type 1 and 3 inflammations, are also orchestrated by acquired and innate immune cells—Th1 and ICL1, in type 1 inflammation, and Th17 and ILC3, in type 3 inflammation. Since our patient showed a low total IgE level (80 IU) with no specific allergens under biologics-free condition [6], eosinophilia could be induced by ILC2-dominant type 2 inflammation.

In the type 2 inflammatory disorders, such as severe eosinophilic asthma, ECRS and EOM, eosinophils are major effector cells [3,17]. Since IL5 is a key cytokine for the differentiation, proliferation, recruitment and activation of eosinophils [12], anti-IL5 monoclonal antibody is a potential option for the treatment for type 2 inflammation [18]. In the previous study, we selected benralizumab, a monoclonal antibody to IL5 receptor α (IL5Rα) on the surface of eosinophils and basophils, which depletes eosinophils and basophils through antibody-dependent cell mediated cytotoxicity [19]. A rapid and remarkable improvement was observed in her asthma control, as well as in her paranasal sinuses and middle ear conditions [6]. After administration of benralizumab, fractional exhaled nitric oxide (FeNO) also returned to normal level (Table 1; November 2018). FeNO primarily reflects IL13, which is released from Th2s, activated ILC2, eosinophils, basophils, mast cells and natural killer T cells [20]. Since activated eosinophils and basophils release IL13, depletion of these cells by benralizumab could reduce IL13, especially in patients with high FeNO level, as in our case [21].

Two types of anti-IL5 antibody, benralizumab and mepolizumab, were used in this patient. Although both antibodies were similarly effective in controlling BA, condition of paranasal sinuses and middle ear gradually deteriorated approximately one year later. As shown in Figure 3A–C, the effectiveness of these two biologics on ECRS was partial, and resulted in the accumulation of fluid in the maxillary sinuses remained (Figure 3B,C). Since depletion of eosinophils either by benralizumab or mepolizumab induced partial clearance of the fluid in the paranasal sinuses and could not prevent the formation of nasal polyps and ear discharge, we speculated that other type 2 cytokines, such as IL4 and 13 could, be involved in the development of ECRS and EOM, in our case.

Since IL4 and IL13 are signature cytokines of type 2 inflammation, dupilumab was used in our case. IL4 binds to type 1 IL4 receptors (IL4R), which is a combination of IL4Rα and the common γ chain, and type 2 IL4R/IL13 receptors which is a combination of IL4Rα and the IL13 receptor Rα1 (IL13Rα1). IL13, on the other hand, binds to IL13Rα1, sharing the same receptor with IL4. A variety of cells, such as eosinophils, mast cells, B and T lymphocytes, smooth muscle cells, goblet cells, fibroblasts and keratinocytes have IL4Rα and IL13Rα1 on their surface [22]. As shown in Figure 3D, fluid of both ethmoidal and maxillary sinuses was promptly and completely disappeared and nasal cavity and eardrum became almost normal after using dupilumab (Figure 2E,F). IL4 and IL13 were reported to inhibit the production of tissue plasminogen activators (tPA) in the polyps of ECRS, and promote fibrin deposition in the tissue due to reduced plasmin activity [23]. A whitish veil of exudate covering the nasal cavity and middle turbinate before using dupilumab, in our case, could be fibrin deposition (Figure 2B), and the inhibition of type 2 inflammation by dupilumab might have contributed to the prompt clearance of the discharge. Since IL13, released either from Th2 in the acquired immune system, or ILC2 and MC in the innate immune system has been reported to stimulate goblet cells [13], another factor for prompt clearance of fluid from nasal cavity and middle ear could be inhibition of IL13 by dupilumab. The best condition of asthma was achieved by dupilumab administration even with eosinophilia and pulmonary infiltration (Table 1; June 2020). Since our patient had a low total IgE level with no specific allergens under a biologics-free condition, acquired immunity through Th2 did not seem to be activated. Therefore, the major source of type 2 cytokines could be derived from ILC2. Since activated ILC2s release a large amount of IL5 and IL13 and smaller amount of IL 4 [14,15], the fact that the best condition of asthma was achieved by dupilumab, even during eosinophilia, suggests that IL13’s induction of airway hyperresponsiveness could be a key cytokine for BA in our case [22].

Although in excellent condition in BA, ECRS and EOM continued using dupilumab, pulmonary infiltration and ground glass opacities with eosinophilia appeared 6 months later (Figure 1A). Pathological examination of biopsy specimen showed hyalinous thickening of the alveolar walls with increased lymphocytes, plasma cells histiocytes and eosinophils. Intra-alveolar buds of granulation tissue were also observed. Since these findings similar to bronchiolitis obliterans were sometimes observed in the chronic eosinophilic pneumonia (CEP) and response to OCS was rapid, diagnosis was likely to be CEP [24,25] As activated eosinophils releases a variety of proinflammatory cationic proteins, cytokines, chemokines and growth factors [26], inflammation and fibrotic processes coexist [27]. The cellular fraction of BALF, in our case, showed that lymphocytes (31%) and histiocytes (58%) dominated over eosinophils (7%), in contrast to the previous report of CEP, which showed pathological findings similar to our case, with a high number eosinophils in peripheral blood and BALF after treatment with dupilumab [28]. CD4/CD8 ratio in our case was very low (0.6), which is the characteristic feature of organizing pneumonia or hypersensitivity pneumonia and is rarely seen in CEP [29].

The patient had no history of eosinophilic pneumonia before the treatment with biologics, despite persistent eosinophilia of approximately 1000–2000/μL (data are not shown). These findings lead us to speculate that the eosinophil activation and transmigration for the development of pulmonary parenchymal inflammation did not occur even in the period of elevated eosinophil count in the peripheral blood. On the other hand, the eosinophil count in peripheral blood temporarily neared zero during benralizumab treatment and increased to a count of 1478/μL during dupilumab treatment. It is possible that acute or subacute eosinophilia, due to a transient increase of eosinophils by dupilumab [30], together with stimulation of the bone marrow by IL-5 released from ILC2 triggered the activation of eosinophils. Since the effect of benralizumab could probably persist at least two months before diminishing, it is possible that the activation and transmigration of eosinophils in the development of pulmonary parenchymal inflammation occurred sometime between 2 and 6 months during the dupilumab treatment. Although it is difficult to discern a clear reason for the development of pulmonary infiltration, the maturation process may be related to the activation and transmigration of eosinophils. IL-4 and IL4Rα are expressed by eosinophil precursors and play a role in the maturation of eosinophils, in cooperation with IL-5 [31]. We hypothesized that the premature eosinophils are more prone to be activated in comparison with mature eosinophils. If this hypothesis is correct, the inhibition of IL4/IL13 by dupilumab could increase the premature eosinophils and develop pulmonary parenchymal inflammation.

Other possible hypothesis for pulmonary infiltration could be the inhibition of the type 2 immune system by dupilumab. Majeski et al. reported that the monoclonal antibody to interferon-γ significantly inhibited the development of fibrosis in a murine model of bronchiolitis obliterans organizing pneumonia and suggested that Th1 plays a key role in fibrotic lesions [32]. Although it is difficult to explain the BALF results, the inhibition of the type 2 immunity by dupilumab might have shifted the immune response toward type 1 inflammation and modulated the inflammatory process to promote profibrotic process, in our case.

In spite of the future possibility of developing CEP, dupilumab was started again by cautiously monitoring peripheral eosinophil count, since nasal and ear symptoms deteriorated QOL. Treatment using both anti-IL4/IL13 and anti-IL5 antibodies together or alternatively may be the best possible treatment modality in the future control of BA with ECRS/EOM, as in our case and as reported previously [33,34]. Since IL13 is probably a key cytokine in our case, anti-IL13 monoclonal antibodies may be the next treatment target, although they are not commercially available for clinical use [35]. Based on the experience of our patient, we summarize a possible treatment algorithm for the management of similar patients (Figure 5).

## 4. Conclusions

Anti-IL5 treatment, either with benralizumab or mepolizumab, was equally effective for controlling BA but less effective in controlling ECRS/EOM in our patient. Dupilumab was, on the other hand, highly effective for controlling bronchial asthma and ECRS/EOM. These results and considerations of cellular interaction in type 2 inflammation support the hypothesis that IL13 may have been a key cytokine in the development intractable BA with ECRS and EOM, in our case. Dupilumab, on the other hand, can induce CEP when eosinophilia occurs. These observations highlight the difficulty in the management of severe asthma with ECRS/EOM using a single monoclonal antibody.

## Figures and Tables

**Figure 1 ijms-22-11209-f001:**
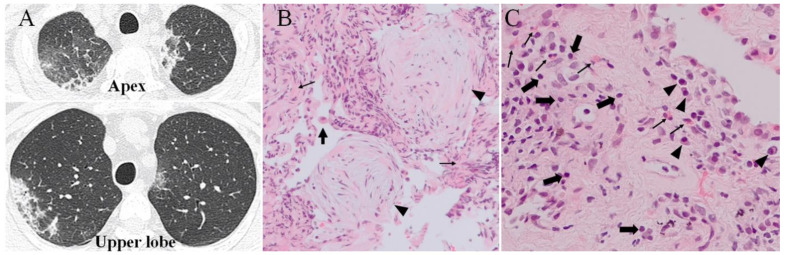
(**A**) Chest computed tomography scan during dupilumab treatment for 6 months. Mosaic patterns of ground glass opacities and consolidations are observed in both the apex and upper lobes. (**B**) Hematoxylin eosin staining of the biopsy specimen (200× magnification). Hyalinous alveolar wall thickening is observed (thin arrows). A portion of the alveolar cavity is filled with granulation tissue composed of fibroblasts, collagen, elastic fibers (arrow heads), and macrophages (thick arrows). (**C**) High magnification (400×). Infiltration of eosinophils (thin arrows), lymphocytes (thick arrows), and plasma cells (arrow heads) are observed.

**Figure 2 ijms-22-11209-f002:**
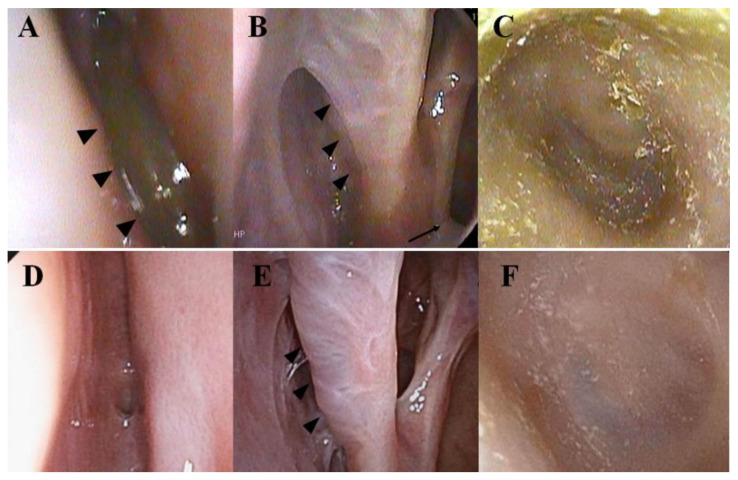
(**A**–**C**) Endoscopic observation of the left nasal cavity and eardrum during mepolizumab treatment for 13 months. (**A**) A nasal polyp in the olfactory cleft (arrow heads). (**B**) The nasal cavity and middle turbinate are edematous and covered with a whitish veil of exudate (arrow heads). Whitish exudate originating from the opening of maxillary sinus is also observed (thin arrow). (**C**) The ear canal and eardrum are covered with a copious amount of dried exudate. (**D**–**F**) Endoscopic observation of the same sites during dupilumab treatment for 1 month. (**D**) No olfactory cleft polyp is observed. (**E**) The middle turbinate appears normal (arrowhead). There is no discharge from the opening of the maxillary sinus. (**F**) The left ear canal and ear drum appear normal.

**Figure 3 ijms-22-11209-f003:**
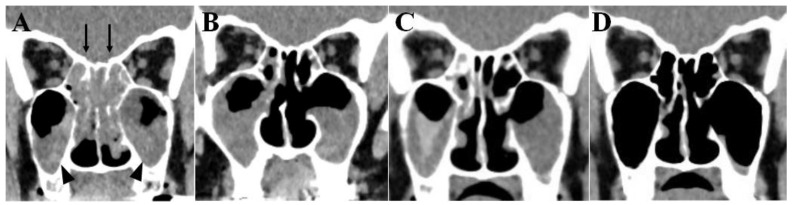
Coronal section of sinus CT. (**A**) Biologics-free condition, adopted from [6] with permission. The bilateral ethmoidal sinuses (arrows) and maxillary sinuses (arrow heads) are filled with fluid. (**B**) At 13 months of treatment with benralizumab. The fluid accumulation in the bilateral ethmoidal sinus is decreased, but unchanged in the bilateral maxillary sinuses. (**C**) At 12 months of treatment with mepolizumab. The fluid accumulation is similar to that observed during the period of benralizumab treatment. (**D**) One month after treatment with dupilumab. The fluid accumulation in both nasal sinuses has almost completely disappeared.

**Figure 4 ijms-22-11209-f004:**
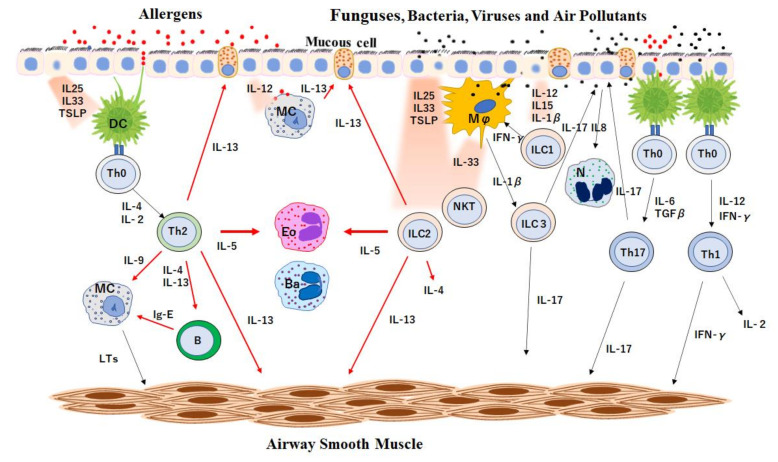
Overview of cell interactions and the cytokine network. DC, MC, Mϕ, Eo, Ba, and N represent dendritic cells, mast cells, macrophages, eosinophils, basophils and neutrophils, respectively. Th0, Th1, Th2 and Th17 cells represent type 0, 1, 2 and 17 helper T cells, respectively. B and NKT represent B lymphocytes and natural killer T cells, respectively. ICL1, ICL2 and ICL3 represent group 1, 2, and 3 innate lymphoid cells, respectively. ILs represent interleukins. TSLP indicates thymic stromal lymphoprotein. LTs represent leukotrienes. MBP, ECP, EDN and EPO represent major basic protein, eosinophilic cationic protein, eosinophil-derived neurotoxin and eosinophil peroxidase, respectively. GM-CSF and TGF-β represent granulocyte-macrophage colony-stimulating factor and transforming growth factor-β, respectively. INF-γ indicates interferon γ.

**Figure 5 ijms-22-11209-f005:**
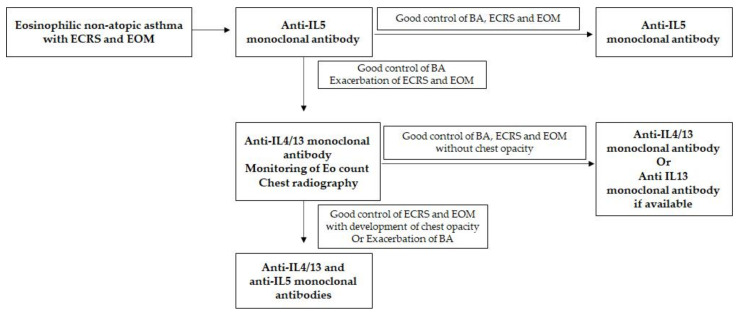
Treatment algorithm for eosinophilic non-atopic asthma with ECRS and EOM. This algorithm is applicable only for those patients who respond to initial treatment with anti-IL5 monoclonal antibody for the control of BA.

**Table 1 ijms-22-11209-t001:** Clinical course of asthma control, quality of life, and laboratory data.

	Biologics Free	2018/Nov 3 Mnths after B	2019/May 10 Moths after B	2019/Oct 15 Months after B	2020/Jan 1 Month after D	2020/June 6 Months after D	2020/Oct 3 Months after M	2021/Jan 6 Months after M	2021/Jun 11 Months after M	2021/Sept 1 Month After D
ACT	14	25	25	25	25	25	24	25	25	25
Mini AQLQ	3.8	6.4	6.8	6.7	6.6	6.5	6.9	6.9	7.0	7
FEV_1_/FEV_1_/FVC %	1.29/72.5	1.87/83.9	1.87/81.3	2.08/83.0	1.87/83.9	1.98/84.6	1.96/85.6	2.04/85.7	2.07/83.8	2.05/86.1
WBC	7300	5700	7900	5600	5600	7500	6400	5000	5500	6700
Eosinophils (%)	12.3	0.1	0.1	0.2	0.1	19.7	2.5	0.8	0.9	0.9
FeNO (ppb)	113	20	21	18	12	25	18	17	20	11
R5	4.26	4.13	5..46	3.01	3.92	3.35	4.62	3.62	3.21	3.15
R20	3.67	3.62	4.79	2.49	3.39	3.72	3.63	2.67	2.63	2.5
Fres	10.3	6.44	6.17	4.64	6.26	6.75	5.98	6.32	6.02	6.4

History of administered biologics. B: benralizumab, D: dupilmab, M: mepolizumab ACT: asthma control test, AQLQ: asthma quality life questionnaire, FEV_1_: forced expiratory volume in one second (L), FEV_1_/FVC%: ration of FEV_1_ vs. forced vital capacity (FVC) expressed by percentage, WBC: white blood cell count, Eosinophils: percentage eosinophils count in WBC, FeNO: fractional exhaled nitric oxide, R5 and R20: respiratory resistance at oscillation frequency of 5 Hz and 20 Hz, Fres: resonance frequency.

## Data Availability

Not applicable.

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
