# Peer review of "IL13 May Play an Important Role in Developing Eosinophilic Chronic Rhinosinusitis and Eosinophilic Otitis Media with Severe Asthma"

_ijms, 2021, doi:10.3390/ijms222011209_

Round 1

Reviewer 1 Report

The authors present a really interesting case report that can be very helpful for the clinicians community, calling into attention possible disease mechanisms. 

I suggest to make the following adressings to improve the manuscript quality:

  1. In Table 1 please correct the double point in R5 at May 2019.
  2. Please Explain why the second time you shifted to mepolizumab instead of benralizumab as before.
  3. Please correct from “dulilumap” to “dupilumab” in the second sentence of the discussion.
  4. Please, as the eosinophilic pneumonia observed is the most interesting part of your article, more details on why you believe that eosinophils raised even though dupilumab was administered is needed, as this drug was able to control their levels the months before, and also, these cells depend on IL-13 secretion. Did you measure cytokine levels in blood or any tissue at this point to see what may be going on?
  5. Related with the previous comment it would be nice to have a hypothesis, and a more insightful comment from the authors regarding which might be causing the chest worsening after the six months of treatment, especially when she was in her best asthma control ever. Have you tested levels of eosinophilic proteins on those tissue samples, or what do you think that might be the origin of the fibrosis if IL-13 is blocked? Maybe an immune polarization towards Th1 response can be considered when IL-13 is blocked. Furthermore, the authors can make suggestions on future possibilities on how to test their hypothesis.
  6. Some kind of general algorithm which describes a possible way of action on this kind of patients might be helpful for the reading clinicians.
  7. Do the authors expect that this patient might develop again the chest abnormalities? How are you planning to follow and control this patient?
  8. Please check through the manuscript that spacing between words and colons are correct.

Author Response

Dear Reviewer 1

Thank you very much for your time of reviewing the revised manuscript above. We have made a possible treatment algorithm (figure 5) for our case along your suggestion. I went through the r manuscript again in this weekend, then felt that I am not quite confident whether it would be better to include algorithm or not, since the development of eosinophilic pneumonia after switching from anti-IL5 antibody to anti-IL4/13 antibody is very rare. Please take account of our concern while reviewing our revised manuscript.

Please accept our deep apology to bother you in this matter, even after sending our response to you.

Reviewer 2 Report

IJMS

Manuscript ID: ijms-1415583
Type of manuscript: Case Report
Title: IL13 may play an important role in developing eosinophilic chronic rhinosinusitis and eosinophilic otitis media with severe asthma.

This article describes the treatment course of a 50-year-old patient with severe bronchial asthma with eosinophilic chronic rhinosinusitis with nasal polyp (ECRS) and eosinophilic otitis media (EOM). This case is interesting, with an excellent response to benralizumab treatment, but with a subsequent complication of ECRS/EOM exacerbation. This required further treatment changes, as described in detail in the text. The article may be interesting and helpful to other clinicians facing similar problems. The authors also discuss the role of interleukin-13 in the development of ECRS/EOM with concomitant severe bronchial asthma.

The article was written in a good language, it is good to read. There may be too much repetition in the discussion from the results section, but on the other hand it makes it easier to read.

I found several typos only to correct:

  • 3, legend to Figure; should be fluid instead of “flu-id:
  • Discussion, line 2: please give the correct names: ”banralizumab” and “dulilumab”
  • Page 7, line 5: should be subepithelial, instead of “subepitherial”
  • Page 7, line 14. There something wrong with the sentence: “In the previous study, we selected benralizumab, a monoclonal antibody to IL5 receptor α (IL5Rα) on the surface of eosinophils and basophils and depletes eosinophils and basophils through antibody-dependent cell mediated cytotoxicity.” I would suggest: There something wrong with the sentence: “In the previous study, we selected benralizumab, a monoclonal antibody to IL5 receptor α (IL5Rα) on the surface of eosinophils and basophils, which depletes eosinophils and basophils through antibody-dependent cell mediated cytotoxicity.”
